# The Genomics of *Streptococcus pneumoniae* Carriage Isolates from UK Children and Their Household Contacts, Pre-PCV7 to Post-PCV13

**DOI:** 10.3390/genes10090687

**Published:** 2019-09-06

**Authors:** Carmen L. Sheppard, Natalie Groves, Nicholas Andrews, David J. Litt, Norman K. Fry, Jo Southern, Elizabeth Miller

**Affiliations:** 1Vaccine Preventable Bacteria Section, Public Health England—National Infection Service, 61 Colindale Avenue, London NW9 5EQ, UK; 2Statistics, Modelling and Economics, Public Health England—National Infection Service, 61 Colindale Avenue, London NW9 5EQ, UK; 3Immunisation and Countermeasures, Public Health England—National Infection Service, 61 Colindale Avenue, London NW9 5EQ, UK

**Keywords:** *Streptococcus pneumoniae*, population, genomics, carriage, epidemiology

## Abstract

We used whole genome sequencing (WGS) analysis to investigate the population structure of 877 *Streptococcus pneumoniae* isolates from five carriage studies from 2002 (*N* = 346), 2010 (*N* = 127), 2013 (*N* = 153), 2016 (*N* = 187) and 2018 (*N* = 64) in UK households which covers the period pre-PCV7 to post-PCV13 implementation. The genomic lineages seen in the population were determined using multi-locus sequence typing (MLST) and PopPUNK (Population Partitioning Using Nucleotide K-mers) which was used for local and global comparisons. A Roary core genome alignment of all the carriage genomes was used to investigate phylogenetic relationships between the lineages. The results showed an influx of previously undetected sequence types after vaccination associated with non-vaccine serotypes. A small number of lineages persisted throughout, associated with both non-vaccine and vaccine types (such as ST199), or that could be an example of serotype switching from vaccine to non-vaccine types (ST177). Serotype 3 persisted throughout the study years, represented by ST180 and Global Pneumococcal Sequencing Cluster (GPSC) 12; the local PopPUNK analysis and core genome maximum likelihood phylogeny separated them into two clades, one of which is only seen in later study years. The genomic data showed that serotype replacement in the carriage studies was mostly due to a change in genotype as well as serotype, but that some important genetic lineages, previously associated with vaccine types, persisted.

## 1. Introduction

Understanding the population structure of pneumococci carried asymptomatically in the nasopharynx is important because carriage is a prerequisite and a reservoir for disease. Characterizing the epidemiology of carriage can help to predict the epidemiology of pneumococcal disease and the effect of vaccination on the pneumococcal population.

The 7-valent pneumococcal vaccine (PCV7) was introduced into the routine infant program in the UK in September 2006, using a 2-, 4-, 13-month schedule and with a catch-up program for children up to 2 years. This was followed by the 13-valent vaccine (PCV13), in April 2010 with the same schedule and no catch-up program. The introduction of PCV7 vaccination resulted in a dramatic reduction in vaccine type invasive pneumococcal disease (IPD) in children and had a profound effect on vaccine type disease in adults in England and Wales [1].

Public Health England (PHE) has undertaken five studies of pneumococcal nasopharyngeal carriage, in 2001–2002 [2], 2008–2009 [3], 2012–2013 [4], 2015–2016 [5] and 2017–2018 [6]. Each of these studies sampled index children and their household contacts using similar study design and geographical locations, and span the time prior to, during and after introduction of pneumococcal conjugate vaccination (PCV) in the United Kingdom (UK).

In each PHE carriage study, there was a similar proportion of participants carrying pneumococcus (24.4–27.7%, 2001–2016 all age groups, all serotypes), but in later carriage studies there was a significant reduction in carriage of PCV7 and PCV13 serotypes to 0% for PCV7 types and 0.7% for PCV13 types in the 2015–2016 study. This reduction was offset by a significant increase in non-vaccine serotypes (NVT) [5]. Although overall incidence of pneumococcal IPD has decreased, a rise in non-vaccine types causing IPD has been a cause for concern in England and Wales [7,8].

To help understand bacterial population evolution and serotype replacement in far greater detail, the use of whole genome sequencing (WGS) data has increased significantly.

One of the first large-scale studies of pneumococcal carriage using WGS data was published in 2014 [9]. This study sequenced over 4300 isolates obtained in a large longitudinal carriage study of mothers and infants on the Thailand–Myanmar border between 2007 and 2010 [10] and highlighted consistency in recombination hotspots between different lineages of pneumococci. The sequence data from this study has been used in genome-wide association studies (GWAS) to create a mathematical model to predict the duration of carriage based on gene content [11] and to test hypotheses about negative frequency dependent selection in maintaining genetic diversity in the pneumococcal population [12].

A study of 516 pneumococcal genomes from carriage isolates obtained from children during the winters between PCV7 vaccine introduction in 2006 and the implementation of PCV13 in 2010 in the UK, found that serotype replacement was accompanied by changes in genotypes and that clonal expansion contributed to serotype replacement [13]. Significant changes in serotype correlated with changes in the dominant sequence type (ST). Associations between ST and serotype were noted in the study, but cases of possible capsular switching or unmasking of previously undetected populations expressing NVT were also identified; as an example of this, ST 199 was associated with both PCV13 serotype 19A and non-vaccine serotype 15B.

In this study, we used WGS data analysis to describe major observations and genomic characteristics in PHE UK carriage isolates from 2001 to 2018 including four years prior to the introduction of PCV7, the period between PCV7 and PCV13 and up to eight years after introduction of the PCV13. The hypothesis was that, in line with data from other studies, changes in serotypes observed in carriage studies were strongly associated with the expansion of distinct clones of pneumococcus in the population rather than large-scale capsular switching from previously seen vaccine-type clones. These data show that, in general, there was an influx of novel genotypes associated with the non-vaccine types, possibly due to unmasking. However, there were cases of capsular switching and/or persistence of clones with non-vaccine type capsular operons that were previously associated more commonly with vaccine types. 

## 2. Materials and Methods 

The PHE carriage swabs from all study periods were obtained from families in the same geographical locations and with similar demographics using the same swabbing techniques and initial isolation methods and therefore provided an ideal set for comparisons. The first PHE carriage study, in 2001–2002, was a longitudinal study (referred to in this paper as “2002”), and sampled children in the Hertfordshire district over 10 months [2]. To select isolates for sequencing from the much larger set available from this study, such that the sample size could be maximized for future analyses, without multiple sampling of the same carriage episode or within-household transmission chain, the two-state model described by Lees et al. [11] for carriage duration, Available online: https://github.com/johnlees/carriage-duration (Accessed on 24 May 2019) was applied to the data. During this analysis, serotypes 15B and 15C were combined into a single type, as they can interconvert during carriage, as were 6A and 6C, which could not be distinguished by serotyping at the time. Successful two-state model fits were achieved for serotypes 19F, 23F, 6B, 14, and 6A/C. All other serotypes used the 19F model, which was conservative with regard to splitting instances, reflecting this serotype’s long carriage duration. If the estimated start and end dates of discrete carriage episodes of the same serotype affected people in the same household, then they were combined as being part of the same transmission chain. The isolate selected for sequencing was the first isolate in each distinct instance of a serotype entering a household, based on these inferred carriage episodes and transmission chains. After this selection, we sequenced 347 isolates from this study. 

For the subsequent cross-sectional studies (where each subject was sampled only once), we obtained WGS data from all the isolates; each study is referred to subsequently by the year in which the study was completed. These studies sampled similar index children and their families in Hertfordshire or Gloucestershire districts as described in their original publications or clinical trial information [3,4,5,6]. For the 2016 and 2018 studies, the isolates had been examined using WGS data during the original study. To complete the set of genomes, we retrospectively sequenced all isolates from the 2009 and 2013 studies. The number of isolates included for the cross-sectional studies were as follows: 2009: 127 isolates, 2013: 154 isolates, 2016: 191 isolates and 2017–2018: 65 isolates.

Sequencing and initial data analysis was performed as described previously [5]. In brief, DNA was extracted using the Qiasymphony DNA Mini kit (Qiagen, Manchester, UK), sequencing was performed using Illumina HiSeq and the Nextera XT kit (Illumina, Great Abingdon, UK) at the PHE Central Sequencing Laboratory. Initial bioinformatic methods included kmerID for species confirmation, Metric Oriented Sequence Typer (MOST) v1.0 for multi-locus sequence typing (MLST) [14] and PneumoCaT v1.0 for serotyping [15]. The kmerID, MOST and PneumoCaT software are available on the PHE GitHub website, Available online: https://github.com/phe-bioinformatics/. WGS data are available at the European Nucleotide Archive (ENA), See Appendix A.

Isolates giving a non-pneumococcal species identification in the kmer ID or a mixed serotype and high non-consensus base metric in the MLST typing indicating a mixed culture were excluded from further analysis. Minimum spanning trees of MLST sequence types (ST) were drawn using Phyloviz [16].

The total number of genomes retained in the analysis was 877 (2002: *N* = 346, 2009: *N* = 127, 2013: *N* = 153. 2016: *N* = 187, 2018: *N* = 64). Isolates that gave a “failed” result on PneumoCaT serotyping and gave a top hit coverage of <25% to any operon sequence in the PneumoCaT capsular type variant database were designated non-typeable (NT).

Serotypes were grouped into the following categories, NT (Non-typeable), NVT (Non-vaccine type), PCV7 (4, 6B, 9V, 14, 18C, 19F, 23F), and PCV13-7 (1, 3, 5, 6A, 19A).

For summary analysis of MLST types and PopPUNK clusters, duplicate examples of the same serotype and ST within the same individual or family were removed to avoid sampling bias due to carriage transmission within close family members or, in the case of the 2002 study, carriage of the same serotype and ST. This removed 135 duplicates and retained 743 genomes. For analysis of the MLST types, any genomes with unassigned novel alleles or STs were removed leaving 727 genomes included (these were retained in PopPUNK summaries).

The data were stratified by period with the categories: pre-PCV7 only (STs appearing in 2002 study only), pre-PCV13 only (2002 and 2009 only) post-PCV7 only (2009 and/or any of 2013–2018 studies), post-PCV13 only (2013 and 2016 or 2018) and persisting STs (2002 and any of the 2013–2018 studies). The genomes with family duplicates or unassigned novel ST/alleles were included in the wider pangenome and phylogenetic analysis, giving a total of 877 genomes.

Changes in the proportion of genomes of each sequence type over the five studies were assessed using Fisher’s exact test, with significance indicated at *p* < 0.01, *p* < 0.001 and *p* < 0.0001. The direction and timing of changes were interpreted from the data. Fisher’s exact test was also used to assess changes in the distribution of clusters within clades over the five studies.

Sequences were assembled de novo using Unicycler v0.4.7 [17]. The resulting contigs were subject to genomic epidemiology analysis using PopPUNK v1.1.6 with the easy-run option and min-kmer 13 [18], and annotated with Prokka v1.13.3 using default settings [19]. Abricate v0.8.13 [20] was used to search for antibiotic resistance (AMR) markers from the genome assemblies using the ResFinder database [21] (updated 23 April 2019 and including 3077 sequences) results are reported where they achieved >99% coverage and identity with no gaps. Any instances of the same resistance marker within isolates from the same household (or individual for the 2002 study) with the same serotype and ST were excluded for the purposes of comparing frequencies.

The general feature format (gff) files produced by Prokka were analyzed using Roary v3.12.0 [22] for core and accessory gene content using the option not to split clusters containing paralogs. A maximum likelihood tree was constructed from the core gene alignment produced by Roary using RAxML (RAxMLHPC 8.2.8, GitHub, Inc., San Francisco, CA, USA) using method “General time Reversible + Optimization of substitution rates + GAMMA model of rate heterogeneity” (GTRGAMMA) [23]. RAxML trees were visualized in R-studio using ggtree [24].

To enable comparison of our carriage genomes to the large collection of genomes available from the Global Pneumococcal Sequencing project (GPS), GPS clusters (GPSC) were assigned using the GPSC reference database and GPSC designations [25], downloaded from the GPS website, https://www.pneumogen.net. PopPUNK [18] was used to assign genomes to GPSCs following the instructions at https://www.pneumogen.net/gps/assigningGPSCs. Clusters obtained from PopPUNK were visualized as networks in Cytoscape using the organic layout algorithm [26,27].

## 3. Results

### 3.1. Serotyping

For the studies in which the original result had been phenotypically derived (2002, 2010 and 2013), the predicted serotype derived using the automated bioinformatical pipeline (PneumoCaT) was compared to the original phenotypical serotyping result for each isolate. Fifty-one discrepancies were noted within the study set. Of these, six were originally resulted as non-typeable but gave a WGS serotype result; 24 were within serogroup discrepancies, eight of which were originally reported as 6A but found to be serotype 6C by WGS. There were a further 21 out-of-serogroup discrepancies, two of which were between serotypes known to be cross reactive (e.g., serotype 29 and serogroup 35). Discrepancies are described in further detail in Appendix A.

As many of the within-serogroup discrepancies were likely to have been due to subtyping errors with the serotyping sera as found in previous studies [15] and we were unable (within the bounds of this study) to fully investigate any discrepancies, the serotypes reported in this study are the result of the WGS analysis throughout.

#### 3.1.1. Population Structure (MLST and PopPUNK)

A total of 150 different MLST sequence types were present in the de-duplicated genomes (not including those with unassigned novel ST/alleles, *N* = 9). A summary of the numbers of genomes by serotype category and study year is shown in Table 1.

Sequence types (STs) persisting throughout the carriage studies and those only appearing in a single period category in relation to vaccine implementation are shown in Table 2.

A minimum spanning tree of the dominant STs in the overall carriage data is shown in Figure 1. The figure shows that only small numbers of related STs were present (as defined by single locus variant), but in some cases the related ST represented both NVT and PCV7 or PCV13-7 serotypes indicating a possible switch in dominant capsular type within the same or a related lineage, e.g., ST 177, 199 and 162. STs seen in all studies were mainly associated with NVT except for these lineages. Figure 1b shows that some of the lineages are only seen in specific time periods of the carriage studies, e.g., ST124 seen only in the 2002 study and ST4149 seen only in 2016 and 2018 studies.

#### 3.1.2. PopPUNK Analysis and Comparison with Global Datasets

PopPUNK analysis on the carriage study data alone produced a total of 81 PopPUNK clusters. When incorporated with the GPSC data, this reduced to 61 clusters, of which 59 corresponded to a known GPSC cluster. There were two local clusters that were not assigned to GPSC, one cluster containing four ST1718, serotype 19F isolates from the 2002 study, later assigned GPSC 931 and one containing one ST9976, serotype 16F from the 2018 study, later assigned GPSC 930.

Figure 2 shows a cytoscape diagram of the local PopPUNK clusters colored by vaccine serotype and by year, labelled by the corresponding GPSC number.

The PopPUNK clusters obtained from the carriage studies show the persistence of several lineages through different time periods, e.g., the local cluster corresponding to GPSC6. Some lineages were associated only with distinct time periods, e.g., clusters associated with PCV7 serotypes (darker green inner fill) from the 2002 study only (yellow outer circle) such as GPSCs 18 and 39, as shown in Figure 2.

When compared to GPSC data, some of the local clusters were shown to be related and placed in a single GPSC; an example of this is serotype 3 isolates which were all a single MLST sequence type (ST180) and fell into two PopPUNK clusters, 27 or 46, in the local dataset analysis. They were placed into GPSC lineage 12 when compared with the global data. GPSCs associated with more than one local PopPUNK cluster are shown in Appendix A.

Notably, some of the GPSC lineages showed time-related changes in serotypes. The three, related local PopPUNK clusters (which were of GPSC lineage 7) shown in Figure 2 at the top left corner were associated with serogroup 23 isolates of different subtypes in different time periods, which were separated into distinct but related clusters in the local dataset. GPSC lineage 4 was associated with 19A, 10A and 15B/C, and GPSC lineage 44 was associated with serotype 19F in earlier years and later 7C and serogroup 24; a slight separation of the VT from NVT within this local PopPUNK lineage can also be seen in the cytoscape diagram (Figure 2).

The frequency and serotype of the top 10 dominant MLST ST seen within each period are shown in Table 3. This table shows the outgoing and incoming sequence types and those that persisted across the time periods and related information such as associated serotypes and GPSC clusters.

Of the dominant sequence types, only ST180, which was solely represented by serotype 3 (PCV13 vaccine type) was divided into separate clusters (27 and 46) in the local dataset PopPUNK analysis (Figure 2). The different clusters suggested a time-related occurrence, with cluster 27 occurring in 2002 (3 of 3 de-duplicated genomes), 2009 (3 of 3), 2013 (1 of 1) and 2016 (1 of 3) and cluster 46 occurring in only in 2016 (2 of 3) and 2018 (1 of 1). This change over the five studies was not, however, significant (*p* = 0.13) due to low numbers.

Table 4 shows the percentage of isolates within each study year that corresponded to each ST, and the STs that have significantly changed over the time period. Eight STs showed a significant increase and six showed significant decreases.

### 3.2. Pangenome Analysis

Summary statistics for all the genomes obtained from the carriage study isolates estimated by Roary are shown in Appendix A. The isolates shared a core genome consisting of 1350 genes (present in ≥99% isolates and with 95% identity). The core genome alignment tree produced by RAxML with a heatmap showing the serotype category, study year of the isolates and some of the detected AMR markers is shown in Figure 3. The nodes on the RAxML tree are colored by GPSC and correspond well with the mid-level clades on the tree.

GPSCs 4, 7 and 44 associated with both VT and NVT are highlighted in the core genome RAxML tree. GPSC 44 contains serotype 19F isolates of ST177 and ST179 with tetracycline resistance marker *tetM* in earlier years, but in later years a separate but related branch of NVT serotypes (serogroup 24 and 7C of ST 177) without the resistance marker significantly increased in number.

GPSC 7 (Figure 2 and Figure 3) includes isolates of serotype 23F, 23B and 23A and shows the switch in dominance of the non-NVT serotypes 23B and 23A in later years. The branch structure of the core genome tree suggests that this is due to replacement of the clade rather than evolution from one serotype to the other. GPSC cluster 4 contains ST199 and related sequence types, and separates to a clade on the core genome tree, represented by serotypes 19A and 15B/C which in turn separate into distinct sub-clades.

GPSC 10 corresponds to a sub-clade of non-typeable isolates which, along with other non-typeable isolates, form a distinct genetic lineage to the encapsulated organisms.

As seen in the local PopPUNK clusters, the serotype 3 isolates (GPSC 12, Figure 3) show two clades on the core genome tree, with isolates in the second, smaller clade corresponding to later studies from 2016 and 2018 only.

Some branches on the core genome tree containing NVT present throughout the five studies (e.g., GPSC 3, Figure 3) expanded post-PCV and show a pattern of branching associated with continuous evolution pre-PCV, then one consistent with rapid clonal expansion in the years post-PCV introduction.

### 3.3. Antibiotic Resistance Markers

Analysis with Abricate detected resistance genes in only 65/877 (7.4%) isolates (all data, not deduplicated). Genes associated with conferring tetracycline (*tetM* or *tet32*), erythromycin (*ermB*), macrolide resistance (*msrD* and *mefA*), aminoglycoside (*aph(3′)-III*) and chloramphenicol (*cat(p194)*) resistance were found in the genomes. After removing replicates within individuals/households, 57 genomes containing resistance markers remained; these are summarized in Table 5. All Abricate results are shown in Appendix A.

Both serotype 3 isolates containing the *tet32* gene were in the same local dataset PopPUNK cluster (46) and from 2016 and 2018; however, *tet32* was not found in the third member of this cluster from 2016.

## 4. Discussion

We describe the genomic characteristics of pneumococcal carriage isolates from children and their household contacts obtained over a 17-year period including the pre- and post-conjugate vaccine era. Nearly all genomes fell within lineages defined by the Global Pneumococcal Sequencing project (GPS), with only two clusters falling outside of previously defined GPSCs (at the time of comparison). The GPSC lineages provided a useful macro-clustering method and generally fit with broad phylogenetic branches on the core genome RAxML tree (Figure 3). Some of the higher resolution phylogenetic clades within the RAxML tree were visible using the local PopPUNK clustering (Figure 2), but grouped together in the GPSC lineages, presumably due to the occurrence of intermediate (ancestral) sequences within the larger GPS dataset. MLST data also provided a useful, and widely recognized, intermediate level of genotype distinction, roughly falling between the local PopPUNK clustering and the RAxML clades in resolution, although in serotype 3, ST 180 the PopPUNK clustering separated isolates into clades whereas MLST type did not.

Statistically significant increases in prevalence of MLST ST 439, 1262, 1635, 2372, 4149, 3811, 62 and 177 in the post-vaccine era were noted. Of these, all but one ST were represented by NVT or non-typeable organisms. The remaining sequence type (ST177), represents a potential serotype switch (Figure 1), seen in two isolates of serotype 19F in the 2002 study and serogroup 24 and serotype 7C in later studies. In the PubMLST (pubmlst.org) dataset, ST177 is only associated with serotype 19F until 2010, when an isolate of serotype 24F appears, it is not associated with 7C until 2013. The rapid increase in ST177 associated with non-vaccine serotypes, particularly serotype 7C, has been noted in invasive disease [28]; the appearance of 7C only post-PCV makes it a rare example of an ST which may indeed have undergone capsular switching in response to vaccine rather than the vaccine unmasking a sub-population which already existed in small numbers pre-PCV.

GPSC 3, represented by serotype 8 and 33F in the global dataset, has been suggested as a possible cause of replacement disease in England and Wales [29]. Serotype 8, the most commonly isolated serotype in IPD in England and Wales in the period 2016–2018 [8], was represented by only four isolates in the data presented here, reflecting their extremely high case-carrier ratio [5], three of which, all from the 2013 study, belonged to GPSC 3. Serotype 33F of GPSC 3 were present throughout the studies, increasing in numbers over time. Our data show a significant increase in ST62, serotype 11A (Table 3 and Table 4), and ST62 genomes were also assigned to GPSC 3 (Figure 3). This supports the suggestion that isolates within GPSC 3 have had a major role in serotype replacement in England and Wales.

The presence of VT and NVT within a single ST was also of particular note in ST199 (associated with serotype 19A and 15B/C) throughout the study periods, showing persistence of 19A, as it also appears in 2016, six years post-PCV13. This possibly reflects the poorer vaccine efficacy against serotype 19A [30]. The branches of 15B/C and 19A, ST199 are distinct in their phylogeny (Figure 3), demonstrating that these lineages were represented by these serotypes throughout the period 2002–2018. ST199 is present in both VT and NVT in the 2002 study, and in serotype 15B/C from as early as 1987 in the PubMLST dataset; therefore, a capsular switching event could have happened in the past, but not as a result of vaccine selection pressure which has simply favored the NVT sub-population of ST199 since introduction.

ST162 was represented in the carriage data in 2002 and 2009 by serotype 19F and later years by serotype 15B/C and serogroup 24 (Table 3). The number of isolates of ST162 decreased after 2002 when the serotypes changed from VT to NVT in the carriage data. ST162 has representatives of 15B/C dating from 2004 and 24 dating from 1998 in the PubMLST data. The appearance of multiple serotypes expressed by isolates of ST162 in the PubMLST data suggests that PCV7 acting on vaccine type variants of this ST led to an expansion of lineage representatives of this important genotype that had an NVT capsule, rather than that the capsule switched in response to vaccination.

In general, the RAxML tree (Figure 3) showed no continuing natural differentiation of branches containing vaccine serotypes after the period in which the vaccine was introduced, because of a loss of the lineage from the data. This indicates that the level at which these lineages were carried in the post-vaccine era is sufficiently low that they were no longer detected in the studies. An exception was GPSC 44 where the probable capsule switch to NVT within ST177 is shown by the clade structure continuing to branch in later years. This supports the hypothesis that most serotype replacement was caused by incoming genetic lineages rather than evolution of existing lineages, with only VT lineages that already had NVT representatives present remaining in the population.

Isolates belonging to GPSC 7 were of note in our carriage data, associated with related lineages of serogroup 23. In 2002, these were dominated by serotype 23F (*N* = 48) with some 23A (*N* = 8), but later by representatives of 23B which appeared from 2009. These lineages form related clusters in the local PopPUNK analysis and clades on the core genome tree. GPSC lineage 7 has been noted as an important lineage in invasive disease in the PCV13 era [29]. A further lineage of 23B isolates, having a genetic subtype of the 23B capsule (designated 23B1) but producing the same polysaccharide as 23B, forms a completely unrelated lineage corresponding to GPSC 5. These have been described previously as an expanding lineage having a separate, unrelated genetic background to 23B [31]. The 23B1 isolates only appear in the carriage data from 2013 and increase in number and proportion (23B/23B1) in 2016 and 2018, with only 23B1 variants being seen in 2018.

Serotype 3 (PCV13 vaccine serotype) persisted at low numbers 2002–2018 despite its inclusion in the PCV13 vaccine. The sequence type (ST180) and GPSC (12) for the serotype 3 isolates did not change throughout the studies (2002–2018). However, a change was noted in the local PopPUNK analysis, where serotype 3 isolates fell into two separate PopPUNK clusters, and this was supported by distinct branches on the maximum likelihood tree (Figure 3, GPSC 12), with one cluster (local cluster 46) associated only with isolates from 2016 and 2018, although this finding was not a statistically significant indication of a shift. Interestingly, three of the serotype 3 isolates from the second clade associated with 2016 and 2018 isolates demonstrated presence of the *tet32* resistance marker.

A study of serotype 3 by Azarian et al. [32], demonstrated the presence of separate clades of serotype 3 within clonal complex CC180 in global data, with a shift in the population structure towards a lineage termed clade II in recent years. Eight of the clade II isolates seen in that study had the *tet32* resistance marker, also seen in our secondary clade and associated with 2016 and 2018 isolates. Although we were not able to fully investigate the clades of serotype 3 that were identified in this study, it is likely that the secondary clade, emerging in the later data, could be equivalent to the clade II serotype 3 from the Azarian et al. [32] study. A shift in clade of serotype 3 was also seen post PCV-13 in carriage data from children in Massachusetts [33]. Further work on this aspect has been undertaken, with reference to invasive isolates of serotype 3 from PHE surveillance.

Few AMR markers were detected among isolates using a basic ResFinder database search on the assembled genomes; however, a decrease in the presence of macrolide resistance markers *mefA* and *msrD*, associated with vaccine serotypes, was detected, as these lineages disappeared from the data after PCV implementation. This change was predominantly serotype 14 (ST 9, GPSC 18), where the decrease in the sequence type was significant (*p* < 0.0001). A subsequent rise in the prevalence of the *tetM* resistance marker was observed, particularly in the non-typeable lineages. The non-typeable isolates were predominantly ST 4149, GPSC 81, where a significant increase was seen in this ST (*p* < 0.0001) post-2013. The effect of vaccination on AMR has been described in previous studies, by elimination of vaccine serotypes carrying AMR resistance reducing overall AMR [34], but also increasing AMR in non-vaccine serotypes [29,35]. No phenotypical antimicrobial resistance testing was performed on the isolates.

In general, an increase in the detection of non-typeable *S. pneumoniae* (particularly ST4149 *p* < 0.0001) was noted; however, this may have been an artefact of differences in isolation rates throughout the studies. In earlier studies, non-typeable (acapsular) pneumococci may have been incorrectly discarded as non-pneumococcal species. The improved accuracy of species identification using WGS in the 2016 and 2018 studies is likely to have increased sensitivity of detection and separation from non-pneumococcal flora. The branch structure of the non-typeable genomes on the core genome tree (Figure 3—GPSC10) shows a gradual branching which might be expected over the time period. This suggests that although increased sensitivity of detection could be supplementing the observed increase, non-typeable lineages may have also emerged and evolved since vaccine introduction as vaccine types were removed from the population. A non-typeable clade associated with AMR resistance marker *tetM* was observed after 2013, delineated from those seen before 2013 on a separate branch on the RAxML tree (Figure 3).

Negative frequency-dependent selection [12] acting on particular rare accessory gene compositions could have determined which clones of NVT filled the niche left by the disappearing vaccine serotypes, and also possibly explain the increase seen in 23B1 variant and secondary serotype 3 clades, but this aspect was not explored in this analysis and further work is required to investigate this.

This manuscript provides a basic genomic description and population structure of carriage isolates obtained from England and has the following limitations; basic assembly-based detection of AMR markers was performed using the ResFinder database only, which could have missed some AMR markers; full analysis of core and accessory genome composition was not attempted and the effect of recombination was not investigated. Further work to fully describe the pangenome characteristics of the genomes in this study could be performed to further understand their evolution.

## 5. Conclusions

The results of our study show that, as seen in other studies [13], replacement serotypes found in the post-PCV studies in England were generally from different genetic backgrounds to the vaccine serotypes. Therefore, the observed population change in carried serotypes over the course of the studies was due to serotype replacement with genotypes present at lower or undetectable frequencies before vaccination becoming more common in the population as vaccine type lineages are removed. However, a small number of important genetic lineages remain that were previously associated with vaccine serotypes and a possible change in the genetic lineage of serotype 3 was noted but was not statistically significant in this study.

## Figures and Tables

**Figure 1 genes-10-00687-f001:**
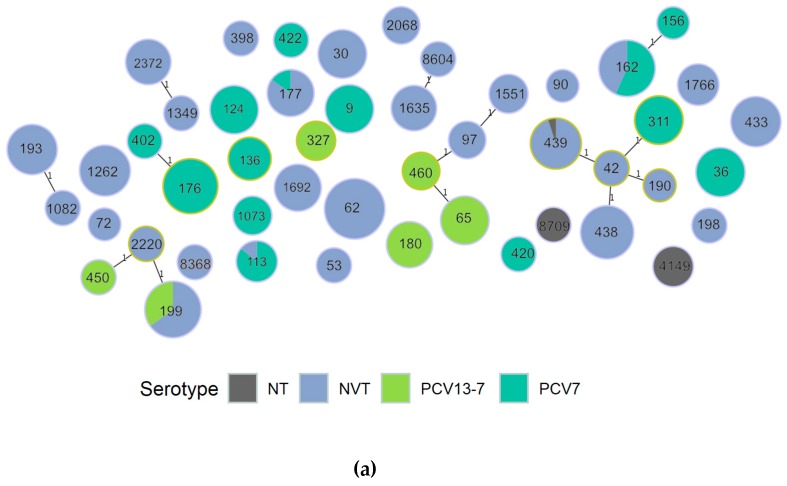
Minimum spanning tree of dominant ST, showing single locus variants and labelled by ST; (**a**) colored by vaccine category for the serotype; (**b**) colored by study year.

**Figure 2 genes-10-00687-f002:**
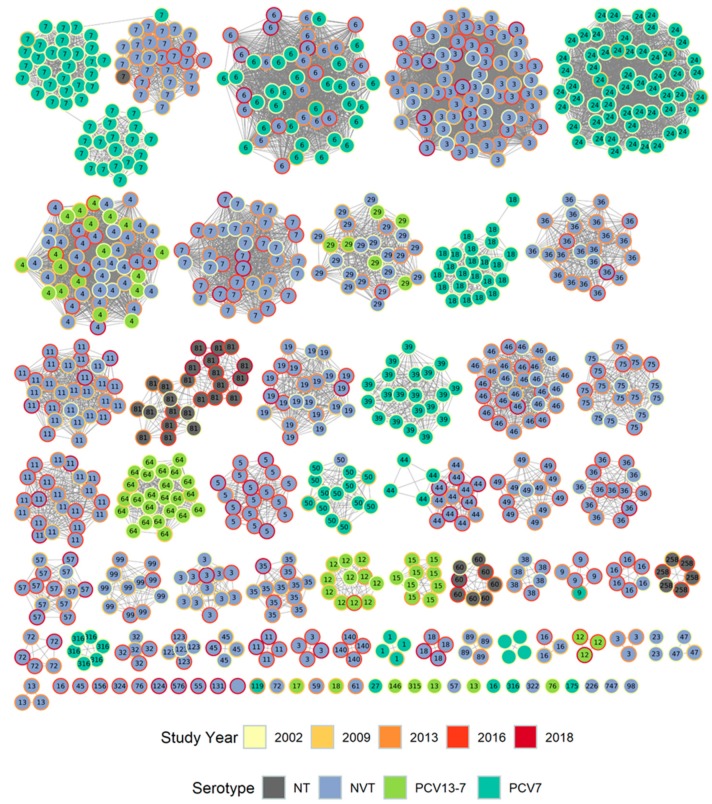
Cytoscape diagram of local PopPUNK clusters colored by vaccine serotype (inner fill) and by year (outer circle), labelled by the corresponding Global Pneumococcal Sequencing Cluster (GPSC) number.

**Figure 3 genes-10-00687-f003:**
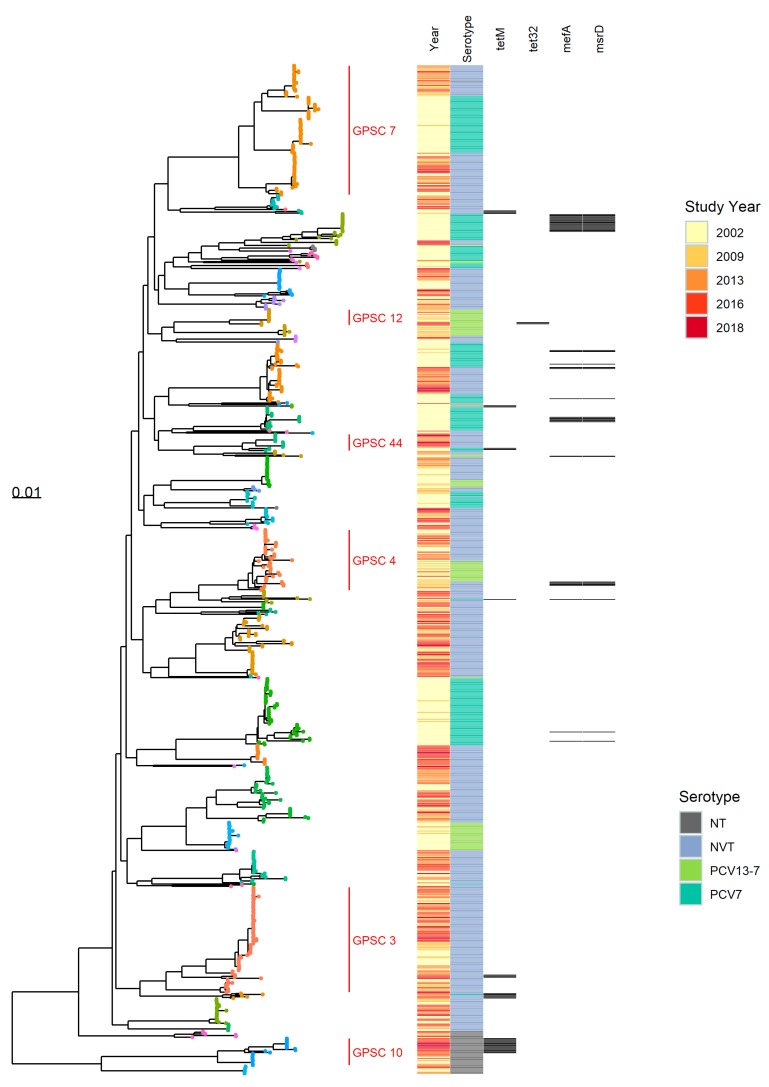
RAxML tree (midpoint rooted) calculated from the Roary core-genome alignment of all carriage study genomes with a heatmap showing study year, serotype category and antibiotic resistance (AMR) markers. Tree tip points are colored by GPSC cluster (key not shown, GPSCs 3, 4, 7, 10, 12 and 44 are labelled). Scale is number of substitutions per site.

**Table 1 genes-10-00687-t001:** Numbers of genomes in each serotype category per study year (deduplicated).

Serotype Category	Number of Genomes per Year of Study					
	2002	2009	2013	2016	2018	Total
PCV7	162	10	2	0	0	174
PCV13-7	36	23	2	5	1	67
NVT	80	66	119	143	48	456
NT	5	1	8	11	5	30
Total	283	100	131	160	54	727

**Table 2 genes-10-00687-t002:** Number of Sequence types (STs) unique to each period of the carriage studies and the number of genomes represented by these STs. The ratio of genomes to STs is included to show the diversity of the genomes.

Time Period	No. STs	No. Genomes	No. of ST Represented by <3 Isolates	Genome to ST Ratio
Persisting Pre- and Post-PCV (2002 and any 2013-2018)	25	343	1	13.7
Pre-PCV7 (2002 only)	50	117	43	2.34
Pre-PCV13 (2002 and 2009 only)	8	87	0	10.9
Post-PCV7 (2009 and any 2013-2018 only)	16	76	7	4.8
Post-PCV13 (2013 and 2016 or 2018 only)	51	104	38	2.0
Total	150	727	89	-

There was a diverse range of STs seen in the carriage studies both pre- and post-PCV vaccine introduction and many STs were represented by only one or two isolates.

**Table 3 genes-10-00687-t003:** Numbers and serotypes of top 10 most frequent ST seen in each period in relation to vaccine implementation, with local PopPUNK cluster and Global Pneumococcal Sequencing Cluster (GPSC) cluster (de-duplicated data).

Period No. ST, No. Genomes	ST	PopPUNK Cluster	GPSC Cluster	Serotype (Number of Genomes)					Total Genomes
				2002 *N* = 283	2009 *N* = 100	2013 *N* = 131	2016 *N* = 160	2018 *N* = 54	
**Pre-PCV7 (2002 only)** **50, 117**	9	13	16	14 (16)					16
	124	16	39	14 (15)					15
	311	1	7	23F (15)					15
	138	3	24	6B (8)					8
	460	11	64	6A (4)					4
	402	3	24	6B (3)					3
	422	33	316	19F (3)					3
	90	51	23	6B (2)					2
	156	5	6	9V (2)					2
	190	6	7	23A (2)					2
**Pre-PCV13 (2002 and 2009 only)** **8, 87**	176	3	24	6B (28)	6B (4)				32
	36	1	7	23F (16)	23F (1)				17
	65	11	64	6A (11)	6A (4)				15
	113	21	50	18C (5) 18B (1)	18C (1)				7
	1073	21	50	18C (4)	18C (1)				5
	327	9	29	6A (3)	6A (2)				5
	420	44	1	19F (2)	19F (1)				3
	450	4	4	19A (1)	19A (2)				3
**Post-PCV7 (2009 and any 2013–2018 only)** **16, 76**	439	1	7		23B (6)	23B (8) NT (1)	23B (5)		20
	1262	15	11		15B/C (3)	15B/C (5)	15B/C (10)	15B/C (2)	20
	1766	22	57		31 (2)	31 (1)	31 (2)	31 (2)	7
	1551	25	35		10A (1)	10A (2)	10A (1)	10A (1)	5
	97	25	35		10A (2)	10A (1)	10A (1)		4
	1082	8	11		21 (3)				3
	8368	26	99		21 (1)	21 (2)			3
	8604	12	36		35F (1)	35F (2)			3
	2220	4	4		15B/C (1)		15B/C (2)		3
	72	47	16		24 * (2)				2
**Post-PCV13 (2013 and 2016 or 2018 only)** **51, 104**	1635	12	36			35F (8)	35F (3)		11
	2372	18	5			23B (1)	23B (6)	23B (4)	11
	4149	10	81				NT (2)	NT (4)	6
	2068	20	36			10A (1)	10A (2)	10A (1)	4
	53	48	3			8 (3)			3
	398	49	13			6C (3)			3
	8709	31	258			NT (1)	NT (2)		3
	198	34	72			35B (1)	35B (1)	35B (1)	3
	1349	18	5				23B (3)		3
	42	6	7				23A (2)	23A (1)	3
**Persisting pre-PCV7 to post-PCV13** **(2002 and any 2013–2018)** **25, 343**	62	2	3	11A (9)	11A (11)	11A (16)	11A (14)	11A (6)	56
	162	5	6	19F (13) 9A (1) 9V (6)	19F (2)	24 * (7) 15B/C (1)	24 * (4) 15B/C (3)		37
	199	4	4	19A (6) 15B/C (10)	19A (5) 15B/C (5)	15B/C (3)	19A (2) 15B/C (5)	15B/C (1)	37
	438	6	7	23A (5)	23A (4)	23A (8)	23A (6)	23A (2)	25
	433	14	19	22F (5)	22F (1)	22F (3) 42(1)	22F (6)	22F (2)	18
	193	8	11	21 (2)	21 (2)	21 (5)	21 (7)	21 (2)	18
	30	7	46	16F (2)	16F (1)	16F (5)	16F (7)		15
	1692	9	29	6C (4)	6C (4)	6C (4)	6C (1)		13
	177	19	44	19F (2)		24 *(3) 7C (1)	24 *(2) 7C (1)	24 *(1) 7C (3)	13
	180	27, 46	12	3 (3)	3 (4)	3 (1)	3 (3)	3 (1)	12

PCV7 serotypes shown in **red**, extra PCV13 serotypes shown in **blue**. NT = Non-typeable. * 24 —serogroup 24 isolates could not be subtyped using the whole genome sequencing (WGS) method.

**Table 4 genes-10-00687-t004:** Sequence types that showed significant increases or decreases over the time period (deduplicated data).

ST	% of Isolates per Study					Exact Test for Change between Years	Comment
	2002 *N* = 283	2009 *N* = 100	2013 *N* = 131	2016 *N* = 160	2018 *N* = 54		
9	5.7					***	Decrease post 2002
124	5.3					***	Decrease post 2002
311	5.3					**	Decrease post 2002
176	9.9	4.0				***	Decrease post 2002
36	5.7	1.0				***	Decrease post 2002
65	3.9	4.0				**	Decrease post 2009
439		6.0	6.9	3.1		***	Increase post 2002
1262		3.0	3.8	6.2	3.7	***	Increase post 2002
1635			7.4	1.9		***	Increase post 2009
2372			0.8	3.8	7.4	***	Increase post 2013
4149				1.3	7.4	***	Increase post 2013
3811					5.6	***	Increase post 2016
62	3.2	11.0	12.2	8.8	11.1	**	Increase post 2009
177	0.7		3.1	1.9	7.4	*	Increase post 2009

*** *p* < 0.0001, ** *p* < 0.001, * *p* < 0.01.

**Table 5 genes-10-00687-t005:** Summary table showing results of Abricate resistance marker detection stratified by year and serotype.

Resistance Marker	No. Genomes	Study Year (No. Genomes)				
		2002 *N* = 346	2009 *N* = 127	2013 *N* = 153	2016 *N* = 187	2018 *N* = 64
*msrD, mefA*	30	14(19), 19F(3), 6A(1), 6B(2), 9V(1)	15B/C(3)		15B/C (1)	
*tetM*	18	19F(1), 6B(2)	6C(1)	19F(1) NT(1)	15A(3), 33F(2), NT(2)	33F(1), NT(4)
*tetM, msrD, mefA, cat(pC194)*	1	19F(1)				
*tetM, cat(pC194)*	2	6B(2)				
*tetM, aph(3′)-III*	2			15A(1)	NT(1)	
*tetM, ermB, aph(3′)-III*	2				NT(2)	
*tet32*	2				3(1)	3(1)
Total	57	32	4	3	12	6

PCV7 serotypes shown in **red**, extra PCV13 serotypes shown in **blue**. NT = Non-typeable N.B. 24—serogroup 24 isolates could not be subtyped using the WGS method.

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
