# Peer review of "The Genomics of Streptococcus pneumoniae Carriage Isolates from UK Children and Their Household Contacts, Pre-PCV7 to Post-PCV13"

_genes, 2019, doi:10.3390/genes10090687_

Round 1

Reviewer 1 Report

This manuscript addresses a different twist to the more common studies of the process of strain replacement post vaccine.  The retrospective use of 4 studies of carriage isolates across important timelines relative to vaccine introductions is appropriate to the question of replacement by new strains vs switching of capsule.  The data supports the replacement by new strains rather than the dominance of capsule switching.  This clarity of conclusion is important to the field.

It would be useful to state categorically if this analysis does or does not support the emerging hypothesis of negative frequency dependent selection.

It would also be very useful to know if the new strains with the new capsules retain major virulence determinants.  There is a prevailing old school idea that new strains of pneumococci are less virulent.  This data suggests that this may not be the case but one way to contribute information in that regard is to look at the core genome not as a whole, but rather look for specific important virulence genes (pneumolysin, choline binding proteins, etc).  Further, this would be very important to any progress in the future for protein based vaccines which assume that new strains will have the key proteins that are considered vaccine candidates

Reviewer 2 Report

I find this an interesting and informative manuscript but needs some minor adjustments. It highlights the complexities of pneumococcal serotype replacement in the PCV era, and the genotypes driving non-vaccine type replacement in carriage in the UK. It relates these findings to observations in disease, impact on AMR prevalence and puts the collection into a global context.

Article: The Genomics of Streptococcus pneumoniae carriage isolates from children and their household contacts, pre-PCV7 to post-PCV13

It may be appropriate and useful to readers to include UK in the title as the sample represents that region and not necessarily Genomics of Streptococcus pneumoniae carriage as a whole.

Abstract

“A Roary core genome of all the carriage genomes was to investigate phylogenetic relationships between the lineages.”

Missing word? “genomes was used to investigate”

“Serotype 3 persisted throughout the study years, represented by ST180 and Global Pneumococcal Sequencing Cluster (GPSC) 12; however, the local PopPUNK analysis and core genome maximum likelihood phylogeny separated them into two clades, one clade only seen in later study years.”

The use of “however” here somewhat suggests the two findings are contradictory rather than hierarchical/complementary. It is not surprising that there is substructure with a GPSC but it is indeed interesting and relevant to your findings.

Introduction

Page 1, line 41 “on vaccine type disease in adults[1].”

As you are only referencing UK data I would adapt the sentence to be more geographically specific.

Page 2 line 52 “for concern England and Wales [7,8].”

Missing word? “for concern in England and Wales [7,8].“

Page 2, line 57 “The sequence data from this study has showed recombination hotspots and has also been used in genome-wide association studies (GWAS) to create a mathematical model to predict the duration of carriage based on gene content [11] and to test hypotheses about negative frequency dependent selection in maintaining genetic diversity in the pneumococcal population [12].”

Something not quite right about how this sentence reads especially “showed recombination hotspots”, it is also quite long.

Page 2, line 77 “cases of capsular switching”

I’m becoming more cautious of phrasing sentences describing capsular switching. I think it is important to make the distinction between capsular switching events that are contemporaneous with PCV use, and those that happened historically and non-vaccine capsular variants are subsequently unmasked.

Materials and methods

Page 3, line 97 “For the subsequent cross-sectional studies”

I understand you subsampled the 2002 study but not the others, as it was a much larger collection. You also attempted to avoid multiple sampling of the same carriage episode for the 2002 sample but not for the remaining studies, for which the motivation and impact on subsequent analysis is less clear.

Page 3, line 111 “(TO BE DONE)”

Needs to be deleted

Results

Page 5, line 181 “post-PCV vaccination introduction“

Likely intended “post-PCV vaccine introduction”

Page 5, line 185 “ST represented both NVT and PCV7 or PCV13-7 serotypes indicating a possible capsular switch within the same or a related lineage e.g., ST 177, 199 and 162.”

Can you give context, have these variants been observed pre-PCV in your collection, or elsewhere e.g. pub MLST or GPS?

Figures 1 and 2.

Apart from descriptions of what these figures represent there appears to be limited references to 1a, 1b or 2 in the results text, particularly regarding what they demonstrate and their interpretation? Regarding clarity, i’m struggling to read the white labels within figure 1 and distinguish the central circle colour from the outer ring in figure 2 particularly for 2002/2009. I also can’t see the GPSC labels the description mentions.

Page 7, line 210-11 “When compared to GPSC data some of these local clusters were combined; an example of this is serotype 3 isolates which were all a single MLST sequence type (ST180) but fell in two PopPUNK clusters, 27 or 46 in the local dataset analysis. They were combined in GPSC lineage 12 when compared with the global data. GPSC associated with more than one local PopPUNK cluster are shown in Supplementary Table 3. Notably, some of the GPSC lineages showed time related changes in serotypes. GPSC lineage 7, was associated with serogroup 23 isolates of different subtypes in different time periods, which were separated into distinct but related clusters in the local dataset. GPSC lineage 4 was associated with 19A, 10A and 15B/C and GPSC lineage 44 was associated with serotype 19F in earlier years and later 7C and serogroup 24”

Really great to see GPSCs and popPUNK used. The objective of the GPSCs was to group related isolates on a global scale into lineages, so that collections can be put into that global context, and to provide groupings that make sense to analyse together e.g. running gubbins, lineage phylogenies. It used the biggest dataset possible so that the missing links between related sub-groups would be captured, much like CCs that can’t be robustly identified using SLV from sparse/limited sampling.

It makes perfect sense to then have other levels of discrimination like you have here, STs or local popPUNK clusters, and the ultimate discrimination would be a tree. I’m being super picky here, but the language (e.g. compared and combined) somewhat suggests there might be some conflict between the different methods but in reality they are complementary and offer different levels of resolution to achieve different aims. For example, it appears that serotype 3 isolates belonged to GPSC12 the major serotype 3 lineage globally, and all of your isolates were ST180 within that, but there was substructure to your isolates which a local PopPUNK run was able to delineate into two distinct but related clusters. It’s a similar story for GPSC4 and 7 for which local popPUNK allowed you to identify sub-clusters that had different temporal distribution. All of which is really interesting! I also used your figure 3 Phylogeny to understand your findings, I think I can spot the subclusters with temporal distribution. If you also annotated the tree with the relevant local popPUNK clusters it would be even more informative regarding these findings. Or alternatively you could always pair all the metadata with the tree on something like microreact and let the eager reader explore.

Table 3.

I think something has gone awry with the formatting, making it quite confusing. in the top row do 9, 13 and 16 refer to ST9, LPPC13 and GPSC16. I can see from ST162 that you can have more than one serotype per ST, for the others though it could be presented as ST9, LPPC13, GPSC16 serotype 14(16/16) to make it much clearer.

“N.B. 23B1 is a genetic subtype of 23B with a different genetic lineage but identical polysaccharide to 23B, serotype 6B/6E represents an isolate that would serotype phenotypically as serotype 6B but has genetic traits of molecular serotype 6E [27].”

Genetic variation within serotypes is interesting and an important area of study. There is great variation in many different serotypes but only 23B1 and 6E highlighted here (https://www.ncbi.nlm.nih.gov/pubmed/31184299). As genetic variation within the CPS does correlate quiet well with the genotypes you are also presenting, I think it would be clearer to just report 23B, and 6B. The subject leads to quite some confusion with anyone less familiar with CPS variation and its impact (or lack thereof) on phenotype!

Table 4.

Is it possible to remind the reader in the figure legend what has been excluded from this analysis, I think it’s the deduplicated data? Again I don’t think any details of Table 4 are specifically mentioned/ referred to in the text?

Page 11 line 249 “6B/6E represents an isolate that would serotype phenotypically as serotype 6B but has genetic traits of molecular serotype 6E [27]”

Again I think you are affording extra genotype information here to one serotype but not the others that isn’t necessary to this particular table.

Page 13, line 278 “ As seen in the local PopPUNK clusters, the serotype 3 isolates (GPSC 12, Figure 3) show two clades on the core genome tree, with isolates in the second, smaller clade corresponding to later studies from 2016 and 2018 only.”

Ok great, I see you come to the GPSC subclades in the context of the phylogeny later in the manuscript, you may wish to include it all in one section? Again the subclades aren’t currently labelled on the tree, adding them may help those with less experience of reading trees. GPSC 10 is also labelled but not discussed in this section and I’d love to know its importance here. Finally what are the colours on the nodes of the tree?

Discussion

Page 13, line 300 “The remaining sequence type (ST177), represents a potential serotype switch, seen in two isolates of serotype 19F in the 2002 study and serogroup 24 and serotype 7C in later studies.”

It might be interesting for the readers to know that this ST expressing serotype 7C was not reported pre-PCV in pubMLST or in the GPS dataset, and therefor this might represent a switch contemporaneous with vaccine use, very few switches have been dated as such.

Page 14, line 315 “From the data available, a capsular switch due to vaccine cannot be proven for this ST due to its presence in both VT and NVT in the 2002 study.”

ST199 expressing 15B/C are also in pubMLST from 1987 (Netherlands) and 1998 (US)….

Page 14, line 320 “ST162 was represented in 2002 and 2019 by serotype 19F and later studies by serotype 15B/C and serogroup 24 and could represent a capsular switching event.”

24F was observed in pubMLST in 1998, the earliest mention of ST 162 and 15B I can find is from 2002 though. Really worth putting your findings into the context of the literature for these potential capsular switches….

Page 14, line 325 “In general, the RAxML tree (Figure 3) showed no evolution of branches containing vaccine serotypes after the period in which the vaccine was introduced, indicating a loss of the lineage from the data. Exceptions were the specific cases of capsular switching described above, where the branch structure indicates continuing evolution.”

I agree with your conclusions here but the use of evolution here is a little non-specific. It’s also not a dated tree so I think it needs rewording along the following lines… “In general, the RAxML tree (Figure 3) showed no evolution (not quite sure what feature you’re referring to here) of branches containing vaccine serotypes isolated after the period in which the vaccine was introduced, indicating a loss of the lineage from the data. Exceptions were the specific cases of capsular switching described above, where the branch structure indicates the continued survival of previously VT lineages.”

Page 14, line 333 “GPSC cluster 7”

GPSC stands for Global pneumococcal sequence cluster, so you have a bit of “cluster cluster” going on here.

Page 14, line 351 “but also increasing AMR in non-vaccine serotypes [33].”

Also reported in ref 29 Lo et al

Page 14, line 352 “No phenotypical antimicrobial resistance testing was performed on the isolates to ascertain their resistance profiles, and therefore these data are an indication of genetic marker presence only.”

I think you actually overstate the valid limitation here, significant work by CDC (PMID:27542334 and subsequent papers) shows resistance genes are a very good predictor of phenotype and ref 24 also does a validation of a number of genetic marker presence.

Page 14, line 360 “The branch structure of the non-typeable genomes on the core genome tree (Figure 3 – GPSC10) shows a gradual branching which might be expected over the time period. This suggests that although increased sensitivity of detection could be supplementing the observed increase, non-typeable lineages have also emerged and evolved since vaccine introduction as vaccine types were removed from the population. A non-typeable clade associated with AMR resistance marker tetM appears to have emerged after 2013, delineated from those seen in before 2013 on a separate branch.”

As it’s not a dated tree (branch length = SNPs not time) you need to be a little careful. Much of the branching structure could be the result of evolution (accumulation of SNPs) that occurred on much larger timescales in a large population of NTs that you have sampled a few of. I expect that if you dated GPSC10 that the time to most common ancestor of many of the NTs would be much larger than the period of time in this study. Given the one estimated mutation rate of 1.57 x10-6 (PMID 21273480) you would expect 72 SNPs could accumulate in a strain over your total study period, but I can’t see a scale bar for this tree. You have very sparse sampling if you consider the total population size of NTs circulating in the UK over this time period. I also can’t see the years of isolation for you NTs in the supplementary, they may not have been detected across the whole period particularly as you acknowledge the early study may not have included them due to similar morphology to non-pneumos. Your tree is also a core gene species wide tree which will have recombining regions, meaning the link between SNPs and time is even more tenuous. Without dating you can’t say when the resistance emerged, only when you first observed it.

Supplementary

I could not find the supplementary materials Table 1 in the files made available to review, that is mentioned to have the ENA accession, so I am unsure of what it contains. A sheet with a row per isolate with ENA accession, year, serotype, ST, GPSC, local popPUNK cluster, AMR profile, inclusion/exclusion from particular analysis would be useful.

Supplementary_Table_2_Serotype_discrepancies sheet 2 column D is reported as excel error “#REF!”. Supplementary Table 5_roary_stats still has tracked changes in it.
